# Vision Transformer Customized for Environment Detection and Collision Prediction to Assist the Visually Impaired

**DOI:** 10.3390/jimaging9080161

**Published:** 2023-08-15

**Authors:** Nasrin Bayat, Jong-Hwan Kim, Renoa Choudhury, Ibrahim F. Kadhim, Zubaidah Al-Mashhadani, Mark Aldritz Dela Virgen, Reuben Latorre, Ricardo De La Paz, Joon-Hyuk Park

**Affiliations:** 1Department of Electrical and Computer Engineering, University of Central Florida, Orlando, FL 32816, USA; nasrinbayat@knights.ucf.edu (N.B.); zubaidah@knights.ucf.edu (Z.A.-M.); reubengel18@knights.ucf.edu (R.L.); 2AI R&D Center, Korea Military Academy, Seoul 01805, Republic of Korea; jonghwan7028@gmail.com; 3Department of Mechanical and Aerospace Engineering, University of Central Florida, Orlando, FL 32816, USA; renoa@knights.ucf.edu (R.C.); ibrahimkadhim@knights.ucf.edu (I.F.K.); mark.aldritz.dela.virgen@knights.ucf.edu (M.A.D.V.); ricardojdelapaz@knights.ucf.edu (R.D.L.P.)

**Keywords:** vision transformer, object detection, collision prediction, visually impaired, self-supervised segmentation, assistive device

## Abstract

This paper presents a system that utilizes vision transformers and multimodal feedback modules to facilitate navigation and collision avoidance for the visually impaired. By implementing vision transformers, the system achieves accurate object detection, enabling the real-time identification of objects in front of the user. Semantic segmentation and the algorithms developed in this work provide a means to generate a trajectory vector of all identified objects from the vision transformer and to detect objects that are likely to intersect with the user’s walking path. Audio and vibrotactile feedback modules are integrated to convey collision warning through multimodal feedback. The dataset used to create the model was captured from both indoor and outdoor settings under different weather conditions at different times across multiple days, resulting in 27,867 photos consisting of 24 different classes. Classification results showed good performance (95% accuracy), supporting the efficacy and reliability of the proposed model. The design and control methods of the multimodal feedback modules for collision warning are also presented, while the experimental validation concerning their usability and efficiency stands as an upcoming endeavor. The demonstrated performance of the vision transformer and the presented algorithms in conjunction with the multimodal feedback modules show promising prospects of its feasibility and applicability for the navigation assistance of individuals with vision impairment.

## 1. Introduction

According to the Centers for Disease Control, there are approximately 12 million people 40 years or older in the United States with some form of vision impairment, including 1 million individuals who are legally blind [1]. The World Health Organization estimates that there are at least 2.2 billion people who experience visual impairment, roughly half of which are either untreated or could have been prevented [2]. Visually impaired individuals (VIIs) face several challenges that require the use of assistive technology (AT) to enable essential activities of daily living. Current ATs targeting VIIs primarily address three major challenges: (i) navigation through indoor and outdoor environments through GPS-determined routes, e.g., [3,4,5,6,7], (ii) obstacle detection via cameras and/or time-of-flight sensors, e.g., [8,9,10,11,12,13,14,15], and (iii) enhancing traditional tools, such as the white cane, to enhance the independence of VIIs, e.g., [16,17,18]. The risk of trips, collisions, or falls increases as VIIs are not able to perceive and recognize objects in their surroundings, particularly their locations, states (static vs. dynamic), and motions (direction and magnitude). Various types of assistive devices have been developed to address these challenges to aid VIIs [19]. For example, enhanced white canes identify large, sound-reflecting objects through echolocation [16]; ioCane [17] and optical pathfinder [18] complement the white cane. However, being close to the ground and short in range, the white cane and other similar technologies fail to offer a comprehensive understanding of the environment and are incapable of informing the user about objects outside of the immediate proximity or objects that are moving at a distance. Another focus of AT is the utilization of body sensors and cameras to collect data about the user’s immediate surroundings and provide feedback to help the user [5,6,7]. The most common applications of navigation aid for VIIs utilize object classification and segmentation via computer vision and deep learning, such as the Blind Sight Navigator [8], Intelligent Belt [9], AngelEye glasses [10], and more [11,12,13,14,15]. Vibrotactile cues have been frequently utilized to inform the user about directional cues, e.g., [3,4]. Despite these advancements and different approaches used, intuitive and efficient ways to convey the contextual information of objects within the environment to provide collision prediction (such as the type of object, its distance and location, and its movement trajectory relative to the user’s current direction) are lacking. To address this gap, we propose an AT system that enhances the situational awareness of VIIs through the utilization of state-of-the-art vision transformers to achieve accurate object classification, an algorithm developed to identify an object’s relative location and movement trajectory for collision prediction, and integrated multi-modality sensory feedback modules to help VIIs interpret their surroundings and objects within. The vision transformer utilized in the proposed object detection architecture offers several advantages over other state-of-the-art models while requiring less computational resources. This is because vision transformers use the mechanism of attention, which weighs the importance of each component of the input data differently [20]. In addition, the presented model deals with the image’s internal structure, using masked image modeling (MIM) [21]. MIM lies in the proper design of the visual tokenizer, which transforms the masked patches into supervisory signals for the target model. We first undertook self-supervised learning with our dataset and then fine-tuned the pre-trained model for two downstream tasks, namely *image classification* and *semantic segmentation*. In related works, extensive research has been conducted on finding objects, such as shops, hotels, restrooms, etc., and detecting obstacles in walking paths [5,19,22,23]; however, a few classes, such as crosswalks, elevators, or room signs, have not been well established [22]. On the other hand, creating and publishing a high-quality image dataset pertaining to VIIs would help advance computer vision-based navigation assistance systems for the visually impaired. Thus, the unique contributions of this work are as follows:The creation and utilization of a dataset that includes object classes that are particularly important for VIIs (e.g., crosswalk, elevator), while accounting for different weather conditions and times of the day (with varying brightness and contrast levels).The successful implementation and demonstration of a state-of-the-art self-supervised learning algorithm that uses vision transformers for object detection, semantic segmentation, and custom algorithms for collision prediction.Multimodal sensory feedback modules to convey environmental information and potential collisions to the user, to provide real-time navigation assistance in both indoor and outdoor environments.

The rest of this paper is structured as follows: In Section 2, we discuss recent related works. In Section 3, we go into detail about the dataset collected for object detection and image segmentation. In Section 4, we explain the methodology of the proposed navigation assistance system and approaches. In Section 5, we describe the evaluation methods. In Section 6, we present the results and discussions. Finally, we conclude the paper in Section 7.

## 2. Related Work

### 2.1. Assistive Systems

In this section, recent works on vision-based navigation and object recognition systems for VIIs are introduced. Computer vision has been the primary method of creating vision-based navigation assistance [24,25,26,27,28], which utilizes cameras and various algorithms to discern different objects. A lightweight convolutional neural network (CNN)-based object identification module was developed for deployment on a smartphone to avoid obstacles and identify nearby objects [29]. Using an RGBD camera, a real-time semantic segmentation algorithm was employed to assist VIIs in maintaining physical distance from others [30]. A DCNN (deep convolutional neural network) for indoor object recognition and a new indoor dataset with 16 object classes were presented [30]. A convolutional neural network (CNN) U-Net image segmentation approach for sidewalk recognition was proposed [31]. A camera and vibration motors positioned on the waist were used to recognize and avoid the obstacle and navigation guidance via vibrotactile feedback [32]. A wearable system with a dual-head transformer for the transparency model was developed, which can partition transparent and general items and carry out real-time pathfinding [33]. Inertial sensors and the smartphone’s built-in camera were used as sensors to guide VIIs in indoor and outdoor areas [34]. A system that uses deep learning and point cloud processing to carry out complex perceptual tasks on a portable, low-cost computing platform was presented [35], which used cutting-edge artificial intelligence (AI) accelerators (Neural Compute Stick-2 (NCS2)), model optimization techniques (OpenVINO and TensorFlow Lite), and smart depth sensors, like the OpenCV AI Kit-Depth, to avoid the need for costly, power-intensive graphical processing unit (GPU)-based hardware required for deep learning algorithms. In-depth analyses of recent navigation systems for VIIs and current challenges are discussed in [19]. A data-driven end-to-end CNN is proposed in [36] to predict a safe and reliable path using RGBD data and a semantic map. To overcome the hardware constraints of using computationally expensive processes, a self-supervised system built on a CNN demonstrated safe and effective navigation assistance with considerably lower processing requirements [37]. An obstacle-detecting method is suggested that uses the modern vision transformer architecture to quickly and precisely detect obstacles [38]. The YOLO-v3 model was used in a device to detect obstacles at the VII’s chest, waist, knee, and foot levels, with auditory feedback to inform the users [39].

### 2.2. Self-Supervised Learning

Self-supervised learning (SSL) can learn discriminative feature representations for image classification, eliminating the requirement for manual annotation on labels. One widely used self-supervised learning is contrastive learning, which compares samples against each other to learn attributes that are common between similar sample pairs and attributes that set apart dissimilar sample pairs [40]. As opposed to contrastive learning, Bootstrap your own latent (BYOL) [41] and Simple Siamese (SimSiam) [42] accomplish similar representation performances without the use of negative sampling [43]. The aim of these models is to maximize the similarity between two augmentations of a single image. A straightforward self-supervised procedure termed DINO was presented in [44], which is interpreted as a type of label-free self-distillation. A different Siamese architecture was suggested, in which one network parameter is updated using the moving average of another network parameter. Using the Road Event Awareness dataset [45], the efficacy of contrastive SSL approaches, such as BYOL and MoCo, was examined [46]. Mask R-CNN [47] was applied to instance segmentation and classification of the images [48]. The colorization task was integrated into BYOL in [49], and the resulting self-supervised method was trained on the cem500k dataset with two different encoders, namely Resnet50 and stand-alone self-attention. IndexNet, a self-supervised dense representation learning method, was used for the semantic segmentation of remote sensing images (RSIs) [50]. IndexNet considers spatial position information between objects, which is critical for the segmentation of RSIs that are characterized by multiple objects. A self-supervised learning approach for the task of automatically generating segmentation labels for driveable areas and road anomalies was presented [51]. The image BERT pre-training with an online tokenizer (iBot) [21] model, enhanced by its pre-training with an online tokenizer, has demonstrated superior performance compared to other existing models. This state-of-the-art approach combines the power of BERT [52], which is a highly effective language model with specialized image understanding capabilities. By incorporating an online tokenizer during the pre-training phase, the iBot model achieved enhanced contextual understanding and representation learning [21].

## 3. Dataset

A total 4 h of video was recorded to build our own dataset from a chest-mounted RGB camera (Kodak PlaySport) while walking in both indoor and outdoor settings within the University of Central Florida’s main campus (Orlando, FL, USA) at various times of the day across multiple days with varying weather conditions, Table 1. The camera was tilted down by 30 degrees to capture the view of the ground as well as the front view of a person. Image frames were then extracted from the collected videos, generating 27,867 photos, resulting in 24 different categories of objects (Figure 1). The size of the images stored in the dataset is 1280 by 720 pixels.

### Data Allocation

Data allocation for training/testing/validation was conducted using an 80/20 split. Moreover, 80% of the available data was allocated for training the model, while the remaining 20% of data was divided between testing and validation. The testing set, comprising 10% of the total data, was used to evaluate the model’s performance and measure its accuracy on unseen examples. Finally, the validation set, which also accounted for 10% of the data, was used to fine-tune the model’s hyperparameters and assess its generalization capabilities.

## 4. Methodology

The methodology employed in this paper is summarized as follows. First, vision transformers were configured to analyze images taken from a body-worn camera, which leverages self-attention mechanisms to capture spatial dependencies and learn important features of the images. Then, Mask RCNN was used to perform object classification, generate masks and bounding boxes, and output object detection and semantic segmentation results. The collision prediction algorithm was developed and implemented, which takes classification and segmentation results as input and computes the centers of the identified objects, determines their relative positions with respect to the user, generates object trajectory vectors, and pinpoints objects that are moving toward the user for collision prediction. The output of this algorithm, specifically the object information in terms of its type, course, and proximity, is conveyed to the user as a collision warning through multimodal feedback that combines auditory and vibrotactile cues. In what follows, the detailed descriptions of the above-mentioned methodologies are presented, including the system configuration and control of the multimodal feedback module.

### 4.1. Framework for Object Detection and Semantic Segmentation

This section provides a detailed description of the presented framework. The vision transformer model for object detection and semantic segmentation is used in the presented framework; Figure 2. The task layers were added to the model that has already been trained in order to perform downstream tasks and adjust the parameters. Vision transformer (ViT) has shown good performance when pre-trained on a large amount of data and applied to several mid- to small-sized image recognition benchmarks (ImageNet, CIFAR-100, VTAB, etc.), which requires less computational resources during training [20]. The MIM task is provided in [53] for pre-training the ViT. Specifically, during the pre-training process, each image contains two views, namely image patches (16 by 16 pixels) and visual tokens (i.e., discrete tokens). The original image is first “tokenized” into visual tokens using a pre-fixed tokenizer $q_{\psi }$. Then, some image patches are fed into the backbone transformer that had been randomly masked. The visual tokens of the original image are predicted by $p_{\theta }$ using the encoding vectors of the masked image, which is the goal of the pre-training.

Self-distillation is a self-supervised discriminative objective presented in DINO (self-distillation with no labels), a new self-supervised system by Facebook AI [44]. In DINO, knowledge is distilled from past iterations of the model itself $\hat{p_{\theta }}$ and not from a pre-fixed tokenizer. Specifically, the two views of each image that are created by applying two random augmentations are put through the teacher–student framework.

iBOT, a self-supervised pre-training framework, is used for feature extraction [21]. iBOT performs masked image modeling (MIM) with self-distillation. First, two augmented views of an input image are generated and named as *a* and *b*. To enable direct image data entry into a standard transformer, the 2D images are divided into $N=h\times w/h_{p}\times w_{p}$ patches, where *h* and *w* are the resolutions of the input images, and $h_{p}$ and $w_{p}$ are the resolutions of the patches [20]. Block-wise masking [53] is performed on *a* and *b*. The minimum number of patches is set to 16 for each block. Block-wise masking is repeated until at least R×N masked patches are generated, where *R* is the masking ratio. The masked views $\hat{a}$ and $\hat{b}$ are generated as Equation (Equation 1)
(1)$$\hat{a_{i}}=a_{i}+m_{i}\left(e_{\left[M\right]}-a_{i}\right),\;i=1,\ldots ,N$$
where $m_{i}$ is a sampled random mask (binary) and $e_{\left[M\right]}$ is a learnable embedding. To conduct the MIM task, the non-masked and masked views are given to the teacher and the student network, respectively. The architectures of both networks are the same, but their parameters vary. The teacher and student networks are defined by a set of weights, $\theta_{t}$ and $\theta_{s}$, respectively, comprising a backbone *f* (vision transformer) and projection heads, $h^{p}$ and $h^{cv}$, where $h^{p}$ is a patch token and $h^{cv}$ is a cross-view token. An exponential moving average (μ) of the student weight is used to update the teacher weights. Specifically, after each training step, Equation (Equation 2) is implemented with μ∈{0,1}.
(2)$$\theta_{t}=\mu \theta_{t}+\left(1-\mu \right)\theta_{s}$$

The teacher and student networks produce probability distributions over *K* dimensions, designated by $p_{\theta_{t}}$ and $p_{\theta_{s}}$, depending on the input image. The teacher network outputs the non-masked views of the *a* and *b* projections of the patch tokens $p_{\theta_{t}}^{p}$, while the student network outputs the masked views of *a* and *b* projections of the patch tokens $p_{\theta_{s}}^{p}$. The original image is restored from the distorted copy by minimizing the average of the loss between $p_{\theta_{t}}^{p}\left(a\right)$ and $p_{\theta_{s}}^{p}\left(\hat{a}\right)$, as well as the loss between $p_{\theta_{t}}^{p}\left(b\right)$ and $p_{\theta_{s}}^{p}\left(\hat{b}\right)$. Most of the models use cross-entropy loss or one of its variants [21]. To improve the model’s functionality, we use focal loss.
(3)$$min\;L_{MIM}=-\sum_{i=1}^{N}m_{i}\left[{\left(1-p_{\theta_{t}}^{p}\left(a_{i}\right)\right)}^{\gamma }logp_{\theta_{s}}^{p}\left(\hat{a_{i}}\right)+{\left(1-p_{\theta_{t}}^{p}\left(b_{i}\right)\right)}^{\gamma }logp_{\theta_{s}}^{p}\left(\hat{b_{i}}\right)\right]$$
where γ is the focusing parameter to be tuned using cross-validation. To distill knowledge from the teacher to the student, *a* is given to $p_{\theta_{t}}^{cv}$ and $\hat{b}$ is given to $p_{\theta_{s}}^{cv}$, then the cross-entropy loss, which shows the similarity between these two projections, is minimized. The same procedure is conducted for *b* and $\hat{a}$, respectively. Similar to the MIM task, these two losses are averaged.
(4)$$min\;L_{CV}=-\sum_{i=1}^{N}m_{i}\left[{\left(1-p_{\theta_{t}}^{p}\left(a_{i}\right)\right)}^{\gamma }logp_{\theta_{s}}^{p}\left(\hat{b_{i}}\right)+{\left(1-p_{\theta_{t}}^{p}\left(b_{i}\right)\right)}^{\gamma }logp_{\theta_{s}}^{p}\left(\hat{a_{i}}\right)\right]$$

Finally, the total loss is calculated as Equation (Equation 5).
(5)$$Loss=L_{CV}+L_{MIM}$$

### 4.2. Collision Prediction Algorithm

Object detection and semantic segmentation results from the vision transformer model are utilized in the collision prediction algorithm, which involves a series of computations to ultimately discern objects that are likely to cause a collision with the user. The algorithm is designed in such a way that it works when the user is either stationary or moving. In what follows, each step-by-step computation is described with an illustration.

(a)Calculation of the center of the objects with respect to the image coordinate frame.

The first step of the algorithm is to calculate the centers of all objects detected and segmented to find their relative location with respect to the user. The camera that captures the front view of the user is mounted on the chest under the sternum; thus, it is reasonable to assume that the image is also center-aligned with the user. The coordinate frame $\left(O_{Global}\right)$ of the image is set at the left bottom corner with its axes defined, as shown in Figure 3. The coordinate frames of segmented objects $\left(O_{I},I=\left[A,B,C,\ldots \right]\right)$ are also defined in the left bottom corners of the bounding box, whose origins are the x and y coordinates that are expressed in the global frame (i.e., $O_{I}=\left[X_{I},Y_{I}\right])$. First, the width and height of the bounding box $\left(w_{I},h_{I}\right)$ and object’s coordinate frame $\left(O_{I}\right)$ are extracted from the real-time image. The center of the object $\left[X_{I_{c}},Y_{I_{c}}\right]$ is computed by half of the width and height of the bounding box $\left(w_{I},h_{I}\right)$ added to the object’s coordinate frame $\left(O_{I}\right)$,
(6)$$\left[x_{I_{c(t_{i})}},y_{I_{c(t_{i})}}\right]=\left[\frac{x_{I_{\left(t_{i}\right)}}+w_{I_{\left(t_{i}\right)}}}{2},\frac{y_{I_{\left(t_{i}\right)}}+h_{I_{\left(t_{i}\right)}}}{2}\right]$$
where I denotes objects (I = A, B, C, …), and subscript $\left(t_{i}\right)$ represents the time frame at which these parameters are computed since the coordinates of $O_{I}$ change over time.

(b)Relative positions of the objects with respect to the user.

Once the center of the objects is known, determining the relative position—left, front, or right—of the objects with respect to the user is straightforward, using the width of the image $\left(w_{img}\right)$.

Case 1: $x_{I_{c(t_{i})}}<\frac{w_{img}}{3}\;\;$ Object I at $t_{i}$ is on the left side of the user.Case 2: $\frac{w_{img}}{3}<x_{I_{c(t_{i})}}<\frac{2w_{img}}{3}\;\;$ Object I at $t_{i}$ is in front of the user.Case 3: $\frac{2w_{img}}{3}<x_{I_{c(t_{i})}}\;\;$ Object I at $t_{i}$ is on the right side of the user.

(c)Identify objects moving closer/further with respect to the user.

There are three possible cases for changes in the proximity of any object with respect to the user, irrespective of their position in the X-Y plane. It can move closer or further if the object is not stationary, or the user can move closer or further if the object is stationary. In either case, such information can be extracted from the changes in the size of the bounding box, as illustrated in Figure 4. First, the size of the bounding box of object I at time i is calculated as $BB_{I(t_{i})}=w_{I(t_{i})}\times h_{I(t_{i})}$, where w and h denote the width and height of the bounding box. Next, $BB_{I}$ is compared between $\left(t_{i}\right)$ and $\left(t_{i+1}\right)$ to yield one of the following three cases:Case 1: $BB_{\left(t_{i}\right)}<BB_{\left(t_{i+1}\right)}$. The Object is moving closer to the user.Case 2: $BB_{\left(t_{i}\right)}>BB_{\left(t_{i+1}\right)}$. The object is moving further from the user.Case 3: $BB_{\left(t_{i}\right)}=BB_{\left(t_{i+1}\right)}$. The object is stationary if the user is stationary, or it is moving in the same direction and speed as the user.
Figure 4(**a**) Identifying objects moving closer/further with respect to the user, (**b**) generating object trajectory vectors.
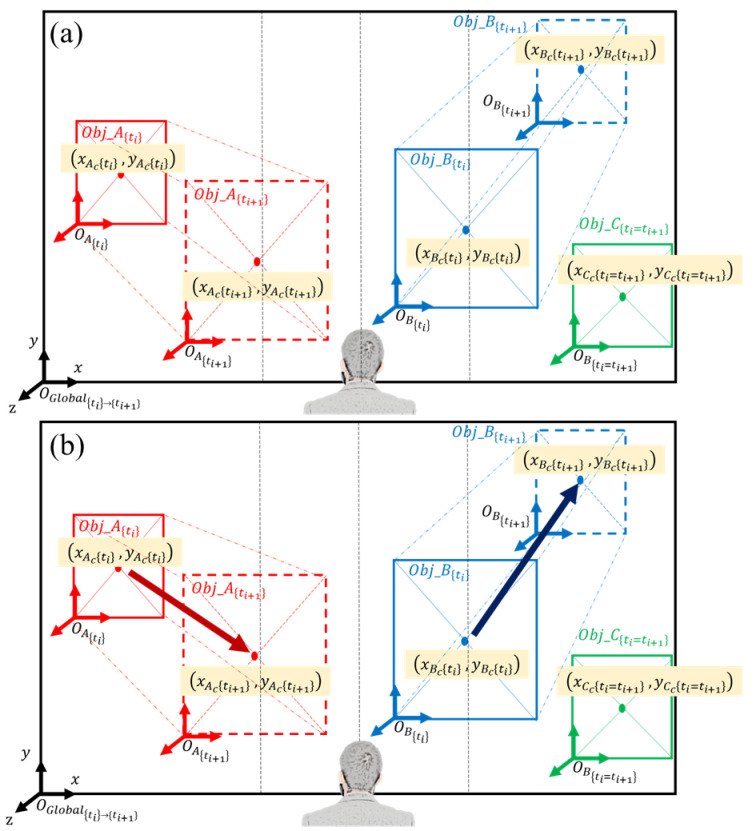



(d)Generate object trajectory vectors.

The following step of the algorithm generates an object trajectory vector using the object center coordinates at $\left(t_{i}\right)$ and $\left(t_{i+1}\right)$. The object trajectory vector expressed at $\left(t_{i+1}\right)$ is $v_{\left(t_{i+1}\right)}=a\hat{I}+b\hat{J}$, where $a=x_{I_{c(t_{i+1})}}-x_{I_{c(t_{i})}}$ and $b=y_{I_{c(t_{i+1})}}-y_{I_{c(t_{i})}}$, as illustrated in Figure 4. If the object center coordinates do not change from $\left(t_{i}\right)$ to $\left(t_{i+1}\right)$, they are the same as in Case 3 of (c) above. The object is stationary if the user is stationary, or it is moving in the same direction and speed as the user. (e) Determine the objects moving toward or away from the user for collision warning.

The final step of the algorithm is to determine the objects that are moving toward or away from the user and identify those that are coming directly toward the user, which would likely collide if they maintain their course. The results of all previous steps of the algorithm are incorporated into this process. The first condition of “moving toward” is step (c) Case 1: $BB_{\left(t_{i}\right)}<BB_{\left(t_{i+1}\right)}$ (i.e., the object is moving closer to the user). Once this condition is met, the relative position of the object (left, front, or right) from step (b) is considered. If the object is on the left side of the user (i.e., $x_{I_{c(t_{i})}}<\frac{w_{img}}{3}$), the signs of a and b of the object trajectory vector from step (d) are evaluated, as shown in Figure 5. If a > 0 and b < 0 (red arrow in Figure 5), it indicates that the object is on the left side of the user, moving closer, and will likely collide. In all other cases (blue arrows in Figure 5), the object is moving closer to the user but its trajectory is less likely to collide; hence, it is “moving away”. The same logic applies to the other two relative position conditions (front and right) in step (b), where there are three cases of “moving toward” when the object is in front of the user, and there is one case of “moving toward” when the object is on the right side of the user. The condition logic for all “moving toward” cases is as follows:(7)$$\left[BB_{\left(t_{i}\right)}<BB_{\left(t_{i+1}\right)}\right]\;AND\left\{\begin{array}{c} \left[x_{I_{c(t_{i})}}<\frac{w_{img}}{3}\;AND\;b<0\right]OR\\ \left[\frac{w_{img}}{3}<x_{I_{c(t_{i})}}<\frac{2w_{img}}{3}\;AND\;b<0\right]OR\\ \left[\frac{2w_{img}}{3}<x_{I_{c(t_{i})}}\;AND\;a<0\;AND\;b<0\right] \end{array}\right\}$$

The integration of auditory feedback (detailed in the next section) to convey the warning to the user is described in Algorithm 1.
**Algorithm 1: Collision avoidance and feedback**1.Capture video from the camera2.fori=0,1,…,N, N=NUMBER OF BOUNDING BOXES IN ONE FRAMEdo3.Store the center point, height, and width of the bounding box in the current frame, and the object trajectory vectors
$\left[x_{I_{c(t_{i})}},y_{I_{c(t_{i})}}\right]=\left[\frac{x_{I_{\left(t_{i}\right)}}+w_{I_{\left(t_{i}\right)}}}{2},\frac{y_{I_{\left(t_{i}\right)}}+h_{I_{\left(t_{i}\right)}}}{2}\right]$$a=x_{I_{c(t_{i+1})}}-x_{I_{c(t_{i})}}$$b=y_{I_{c(t_{i+1})}}-y_{I_{c(t_{i})}}$4.Check if the object is moving toward the user:a.$\mathrm{If}\;w_{I_{\left(t+1_{i}\right)}}\times h_{I_{\left(t+1_{i}\right)}}>w_{I_{\left(t_{i}\right)}}\times h_{I_{\left(t_{i}\right)}}\;\mathrm{then}$$\mathrm{If}\;x_{I_{c(t_{i})}}<\frac{w_{img}}{3}\;AND\;b<0$print “Name of the detected object” is approaching you from your left side!$\mathrm{elseIf}\;\frac{w_{img}}{3}<x_{I_{c(t_{i})}}<\frac{2w_{img}}{3}\;AND\;b<0$print “Name of the detected object” is in front of you!$\mathrm{elseIf}\;\frac{2w_{img}}{3}<x_{I_{c(t_{i})}}\;AND\;a<0\;AND\;b<0$print “Name of the detected object” is approaching you from your right side!b.$\mathrm{If}\;w_{I_{\left(t+1_{i}\right)}}\times h_{I_{\left(t+1_{i}\right)}}=w_{I_{\left(t_{i}\right)}}\times h_{I_{\left(t_{i}\right)}}\;\mathrm{then}$print “Object is static!”c.end5.Store the current center points and distance for the next iteration$\left[x_{I_{c(t_{i})}},y_{I_{c(t_{i})}}\right]=\left[x_{I_{c(t+1_{i})}},y_{I_{c(t+1_{i})}}\right]$6.end

### 4.3. Auditory and Vibrotactile Feedback

Two different modalities of sensory feedback, auditory and vibrotactile, are utilized to inform the user of a potential collision with surrounding objects. For auditory feedback, the collision prediction algorithm’s outputs—“[object I] moving toward the user from [position (left, front, or right)”]—are transformed into speech through Festvox [54], an offline software offering text-to-speech conversion, which supports multilingual speech synthesis on multiple platforms. Then the auditory feedback is sent to the user via Bluetooth earbuds. Simultaneously, vibrotactile feedback is implemented to inform the user about the direction and speed of an object moving toward the user; Figure 6. This is achieved by using either the wrist or waist interface integrated with vibration motors, controlled by a microcontroller (Adafruit nRF52840 Feather Sense) connected to a Servo Driver (PCA9685). The Feather Sense contains an inertial measurement unit (IMU), which provides orientation information. The servo driver accommodates up to eight vibrotactile motors, each wired to a BJT NPN transistor to control the voltage applied to the motor. IMU data are periodically sent to the main computer (1 Hz) to determine the body segment orientation relative to the global coordinate frame (earth), such that the correct vibrotactile motor is turned on to indicate the direction, irrespective of the wrist/waist orientation. Based on the data collected by the computer, commands are sent to the microcontroller via Bluetooth Low Energy (BLE) to control specific motor(s) to vibrate, and the strength and duration of the vibration. The strength of the vibration is controlled by the duty cycle of the PWM signal with 16-bit resolution.

## 5. Evaluation Setting

In this section, general evaluation settings, e.g., pre-processing steps, implementation specifics, and evaluation metrics are described in detail.

### 5.1. Networks Architecture

A backbone *f*(ViT) and projection head *h* make up the teacher and student networks. The outputs of the backbone *f* are used as features for object detection and semantic segmentation tasks. The projection head is constructed using a three-layer multi-layer perceptron (MLP) with hidden dimensions of 2048. This is followed by L2 normalization and a weight-normalized fully connected layer with K dimensions [44]. Vision transformers are used as backbone *f* with a patch size equal to 16×16. The total number of patch tokens corresponds to the ratio of the photo size used for both pre-training and fine-tuning (224 by 224) to the patch token size (16 by 16), resulting in 196 patch tokens. The ViT model sizes are denoted as base, large, or huge, which determine the number of layers, heads, parameters, and the MLP size, as explained in [20]. In this work, the base ViT with a patch size of 16 is employed for model pretraining.

### 5.2. Implementation Specifics

All the models were implemented using the PyTorch machine learning library, with a learning rate of $10^{-6}$, and 0.4 weight decay. The model was pre-trained using the AdamW [55] optimizer, with a batch size of 1024 on the ImageNet-1K [56] training set. The model was fine-tuned using our dataset with 1000 epochs, a learning rate (lr) of $10^{-6}$, weight decay (wd) of 0.4, accuracy (ACC) of 92%, and base ViT with a patch size of 16 as the backbone. A Lambda Quad deep learning workstation was employed to conduct the experiments. The machine was equipped using the Ubuntu 20.04.3 LTS operating system, Intel Core™ i7-6850K CPU, 64 GB DDR4 RAM, and 4 NVIDIA GeForce GTX 1080 Ti graphics processing units (GPUs).

## 6. Results and Discussion

The presented model’s attention maps are presented in Figure 7. One of the fundamental blocks of the transformer is self-attention. This computational primitive helps a network learn the hierarchies and alignments existing in the input data by quantifying paired entity interactions. For vision networks to acquire greater robustness, attention is a crucial component [20]. By providing more attentively visualized results to each part of the image, our model demonstrated its capability to distinguish multiple objects or different parts of a single object.

### 6.1. Object Detection and Semantic Segmentation

We used Mask RCNN [47] to perform both classification and object location tasks at the same time by generating masks and bounding boxes simultaneously. Object detection and semantic segmentation were performed on COCO [57] and ADE20K [58] datasets, respectively. Figure 8 shows an example of object detection and semantic segmentation results. The inability to successfully detect the bicycle rack in Figure 8 could be attributed to a lack of diversity in the dataset used for training the model. If the model was trained on a limited set of bicycle rack images that do not cover various angles, lighting conditions, and designs, it may struggle to detect an object that is partially blocked or overlapped by other objects or showing from an angle at which the key features are not visible.

For a thorough grasp of a scene, object detection alone is insufficient, especially when it comes to identifying sidewalks, roadways, crosswalks, and vegetation. Semantic image segmentation models were employed to facilitate safe navigation of VIIs through the acquisition of semantic knowledge of the front scene. The model must first be constructed, and the pre-trained weights must be loaded. Semantic segmentation can be viewed as a classification problem at the pixel level; linear head and Mask RCNN were used for the task layer.

Classification results on the validation dataset are presented in Table 2 for each of the 24 classes included in the model. The precision metric measures the ability of the model to correctly identify instances of a particular class. The precision scores for most classes are above 0.9; this indicates that the model has good precision. The recall metric represents the ability of the model to correctly identify all instances of a particular class. The recall scores for most classes are also high, ranging from 0.85 to 1.00, indicating that the model has high sensitivity in detecting instances of each class. The F1 score is the harmonic mean of precision and recall, providing a balanced measure of a model’s performance. The F1 scores for most classes are also high, ranging from 0.80 to 0.97, indicating an overall good balance between precision and recall. The support column in Table 2 shows the number of instances in each class used in validation. The overall classification accuracy of the model on the entire dataset is 0.95, meaning that it correctly classifies 95% of the instances. The confusion matrix from the classification of 24 classes on the validation dataset is provided in Appendix A. The number of samples used in the validation varies across different classes. The effect of such an unbalanced dataset on the classification results leads to low precision, recall, and F1 scores for the minority classes, while the performance on the majority classes may still be relatively high. Our model implements the transfer learning technique on the pre-trained model, which can effectively address this issue of imbalanced data, as evidenced by the reasonable precision, recall, and F1 score of a minority class in our validation dataset (e.g., trash bin). In summary, the classification model demonstrates good performance across most classes.

### 6.2. Collision Prediction

The collision prediction algorithm was tested on sample images in which a person was walking toward the user; Figure 9. It can be seen that the algorithm calculates and compares the center of the bounding box of the person, then creates a movement trajectory vector, along with distance changes between two different time frames. It demonstrates that the algorithm properly functions and outputs the expected result. The model shows robustness when dealing with objects of different sizes, indicating its ability to handle diverse scale variations effectively.

### 6.3. Limitations and Future Work

The completion time of object classification and semantic segmentation of the implemented model is less than 0.5 s, and with the collision prediction algorithm running in tandem, the total response time is estimated to be 0.8 s. Visually impaired individuals rely heavily on accurate and reliable environmental perceptions to avoid obstacles, hazards, or potential dangers. Therefore, 95% accuracy of the presented classifier achieved may still not be good enough to ensure the safety of visually impaired individuals when navigating dynamically changing environments. Moreover, any errors in the semantic image segmentation model could also lead to critical consequences, especially in outdoor settings where uncertainty and complexity are higher. To address this, further optimization and fine-tuning of the model, as well as incorporating additional techniques, will be sought to further enhance the robustness and accuracy of the presented model, such as simultaneously running multiple semantic segmentation models for cross-referencing the segmentation results.

## 7. Conclusions

This work presents the implementation and evaluation of vision transformer-based object classification, a custom-built algorithm for collision prediction, and the integration of multimodal sensory feedback to provide real-time feedback to VIIs to help carry out indoor and outdoor navigation safely. The dataset used in the developed model was collected from indoor and outdoor settings under different weather and time-of-day conditions, generating 27,867 photos with 24 classes, which are accessible to the AT research communities. In addition, the developed model includes a few classes that have been less focused on in previous works, such as elevators. The collision prediction algorithm analyzes changes in object locations between frames, allowing the system to predict whether the object is currently in motion or stationary, their estimated trajectory, how far away they are from the user at any given moment, and their current movement direction, with respect to the user. These data are provided to the user through a combination of vibrations and auditory cues to allow VIIs to make informed decisions about how to interact with their surroundings. Future work will include (i) testing and evaluation of the feedback module to assess its functionality and usability, (ii) migrating the model and algorithms to a compact, high-performance single board computer to enhance portability, and (iii) conducting human subject trials with the developed system to validate the overall system performance in both indoor and outdoor navigation scenarios.

## Figures and Tables

**Figure 1 jimaging-09-00161-f001:**
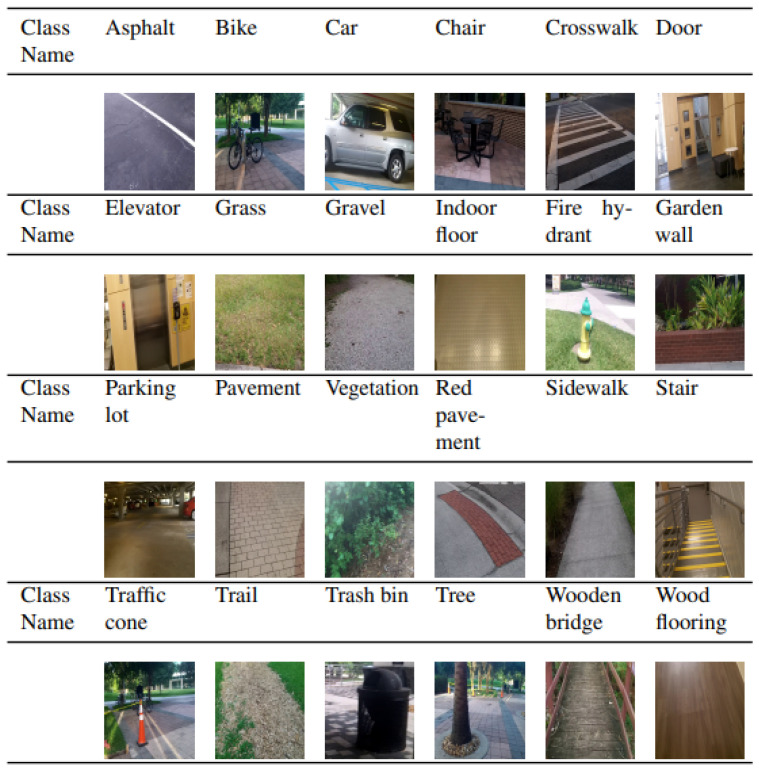
Representative images of 24 classes determined from the collected dataset.

**Figure 2 jimaging-09-00161-f002:**
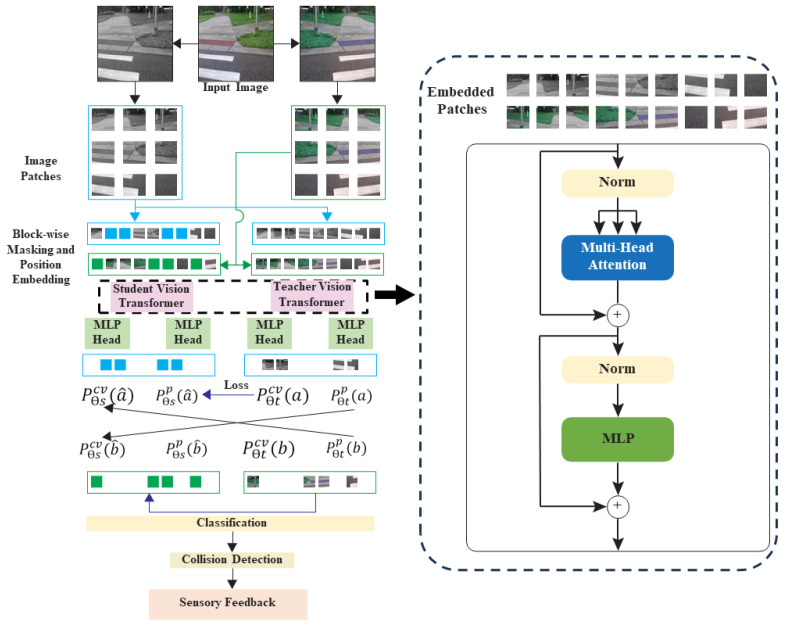
Overall framework of the vision transformer-based object classification and feedback.

**Figure 3 jimaging-09-00161-f003:**
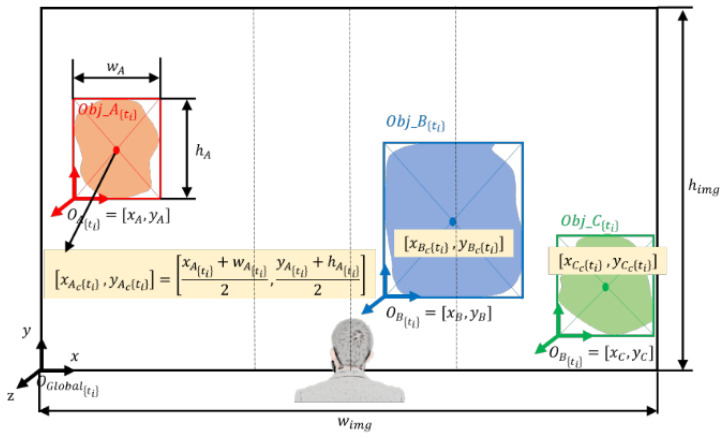
Calculation of the centers of the objects and their relative positions with respect to the user.

**Figure 5 jimaging-09-00161-f005:**
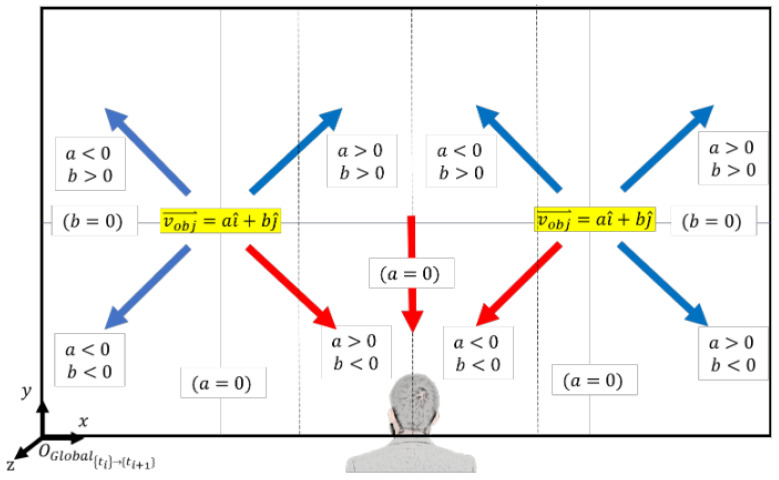
Objects moving toward/away for collision warning.

**Figure 6 jimaging-09-00161-f006:**
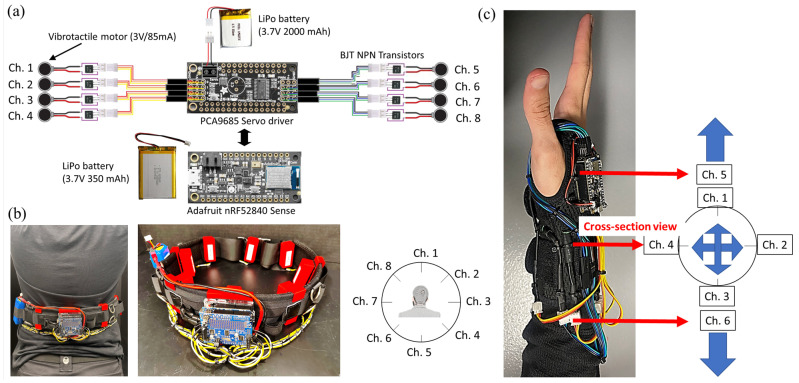
Vibrotactile feedback module: (**a**) electronic hardware schematics, (**b**) waist interface, and (**c**) wrist interface.

**Figure 7 jimaging-09-00161-f007:**
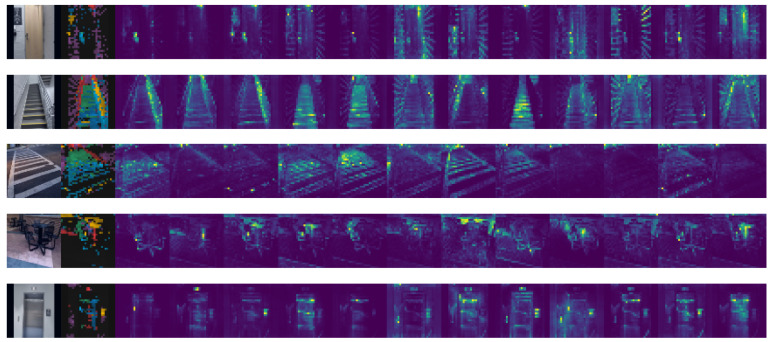
Visualization of the attention maps of 12 heads of the final layer of ViT-B/16, with no supervision.

**Figure 8 jimaging-09-00161-f008:**
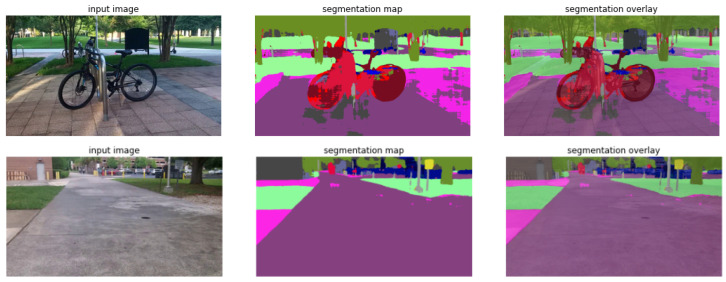
Examples of object detection and semantic segmentation performed by the proposed framework.

**Figure 9 jimaging-09-00161-f009:**
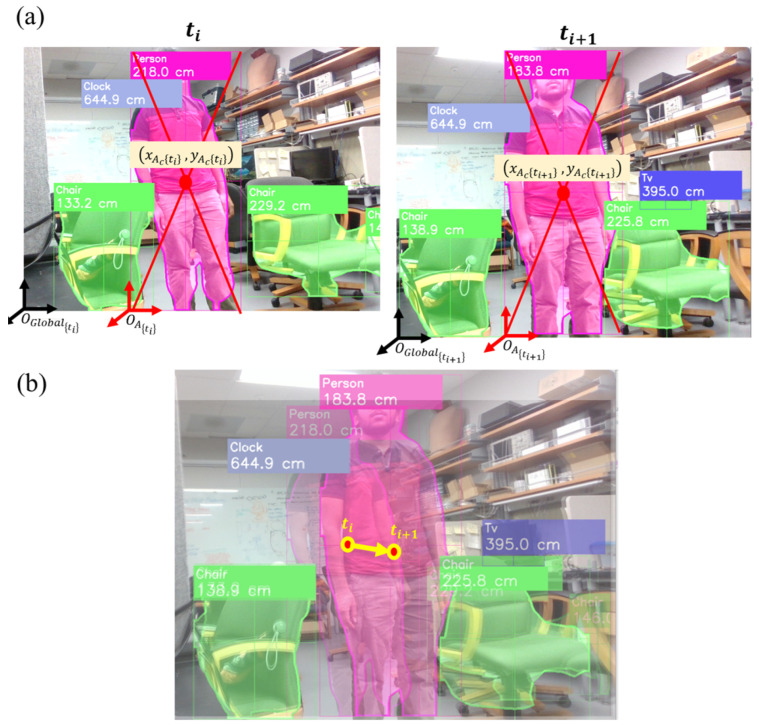
Demonstration of the implemented algorithm: (**a**) bounding box and the center of objects at $\left(t_{i}\right)$ and $\left(t_{i+1}\right)$, and (**b**) generation of object trajectory vector.

**Table 1 jimaging-09-00161-t001:** Date, time, and weather conditions of multi-day data collection.

Date	Time of the Day	Weather
11 May 2022	5 pm	sunny
13 May 2022	4 pm	sunny
16 May 2022	11 am	sunny
20 May 2022	6 pm	cloudy
23 May 2022	8 pm	cloudy
24 May 2022	10 am	sunny
24 May 2022	7 pm	sunny
25 May 2022	3 pm	sunny
26 May 2022	11 am	sunny
28 May 2022	6 pm	sunny
29 May 2022	3 pm	sunny

**Table 2 jimaging-09-00161-t002:** Classwise classification results on the validation dataset.

Class Number	Class Name	Total Number of Images	Precision	Recall	F1 Score
0	Asphalt	1076	0.97	0.97	0.97
1	Bike	62	0.90	0.90	0.90
2	Car	280	0.92	1.00	0.96
3	Chair	507	0.98	0.97	0.97
4	Crosswalk	3524	0.94	0.91	0.93
5	Door	1829	0.94	0.94	0.94
6	Elevator	347	0.96	0.96	0.96
7	Grass	569	0.98	0.97	0.98
8	Gravel	1296	0.93	0.95	0.94
9	Indoorfloor	758	0.96	0.96	0.96
10	Fire hydrant	430	0.96	0.93	0.94
11	Garden wall	95	0.93	0.93	0.93
12	Parking	965	0.98	0.95	0.96
13	Pavement	1111	0.93	0.93	0.93
14	Vegetation	732	0.93	0.96	0.94
15	Red pavement	629	0.97	0.95	0.96
16	Sidewalk	2732	0.91	0.96	0.93
17	Stair	4128	0.96	0.96	0.96
18	Traffic cone	70	0.97	0.85	0.90
19	Trail	3651	0.94	0.96	0.95
20	Trash bin	101	0.67	1.00	0.80
21	Tree	1690	0.92	0.94	0.93
22	Wooden bridge	1399	0.96	0.96	0.96
23	Wood flooring	420	0.97	0.91	0.94
Accuracy					0.95

## Data Availability

The utilized dataset in this study is publicly available at https://drive.google.com/drive/folders/1ecBLxPrpZOErCwNGVL4JKeDGHGi4R0Dd?usp=share_link (accessed on 1 June 2023).

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
