# Peer review of "Vision Transformer Customized for Environment Detection and Collision Prediction to Assist the Visually Impaired"

_2313-433X, 2023, doi:10.3390/jimaging9080161_

Round 1
Reviewer 1 Report
The manuscript presents the implementation and evaluation of a vision transformers-based object classification, a custom-built algorithm for collision prediction, and the integration of multimodal sensory feedback to provide real-time feedback to visually impaired individuals to assist with safe indoor and outdoor navigation.
However, it is hard to agree with the method of classifying objects and obtaining partition information through transformers to determine whether an object will collide, and providing multisensory feedback to visually impaired people. Using a visual information object detector, we can obtain location information of the object. Therefore, the location and classification information from object detection not only aids in collision avoidance but also makes it easier to guide to the destination.
Furthermore, while a 95% classifier performance may be higher than other classifiers, it might not be a good enough performance to ensure that visually impaired individuals can function safely outdoors without facing danger.
Author Response
The manuscript presents the implementation and evaluation of a vision transformers-based object classification, a custom-built algorithm for collision prediction, and the integration of multimodal sensory feedback to provide real-time feedback to visually impaired individuals to assist with safe indoor and outdoor navigation.
However, it is hard to agree with the method of classifying objects and obtaining partition information through transformers to determine whether an object will collide and providing multisensory feedback to visually impaired people. Using a visual information object detector, we can obtain location information of the object. Therefore, the location and classification information from object detection not only aids in collision avoidance but also makes it easier to guide to the destination.
⦁ Thank you for the comment. The vision transformer is used for object classification, and segmentation not collision prediction. As illustrated in the overall framework (Figure 2), the collision prediction algorithm (section 4.2) utilizes the output of the vision transformer (attention maps) and Mask-RCNN (Semantic segmentation) (detailed in 6.1). Once the bounding boxes are drawn from images, this algorithm executes a series of computations to predict which objects will have a high probability of colliding with the user.
Furthermore, while a 95% classifier performance may be higher than other classifiers, it might not be a good enough performance to ensure that visually impaired individuals can function safely outdoors without facing danger.
⦁ Thank you for your feedback. Achieving a 95% classification performance is commendable and indeed is higher than other classifiers. However, when assisting visually impaired individuals navigating outdoors, safety is of utmost importance. For this specific application, even this classification accuracy, as you correctly noted, may not be good enough. The following paragraph has been added to point out your feedback in the discussion section.
⦁ 6.3. Limitations and Future Work (pg 14)
“Visually impaired individuals rely heavily on accurate and reliable environmental perception to avoid obstacles, hazards, or potential dangers. Therefore, 95% accuracy of the presented classifier achieved may still not be good enough to ensure the safety of visually impaired individuals when navigating dynamically changing environments. Moreover, any errors in the semantic image segmentation model could also lead to critical consequences, especially in outdoor settings where uncertainty and complexity are higher.To address this, further optimization and fine-tuning of the model as well as incorporating additional techniques will be sought to further enhance the robustness and accuracy of the presented model, such as simultaneously running multiple semantic segmentation models for cross-referencing the segmentation results. “
Reviewer 2 Report
This paper presents a vison guided navigation system supposed to help implementing collision avoidance for visually impaired humans. The system uses a Mask-RCNN to perform object classification. A self-supervised teacher-student framework is used for feature extraction. Collision Prediction Algorithm is developed in following stage use the obtained object classification results as input. Above approaches are demonstrated using a body worn camera and guide the human operator through a wrist interface. The topic is interesting, and the results is promising. Although I am not convinced that VII guidance is the best application of this technology and there are still some practical issues need to be overcome, the potential of this technology, in both academic research and industrial usage, is huge. Expected few critical information missing, the manuscript is well prepared. I would like to support the publication after following information are provided.
1. In section 4.2. line 257. The author calculate the bounding box of the object . However, it is not clear to me how they get the initial size of the object w_I and h_I. Are they extracted from the real-time image or estimated from pre-trained object database?
2. As the proposed method is claimed as "enabling real-time identification", the speed or average time (at least estimated response time) of the algorithm to finish the task should be given.
3. In figure 8. i guess the bicycle rack is not able to be detected successfully, is it? It does not affect the novelty and efficiency of the method proposed. However, it maybe can be considered a limitation of current version of the algorithm or interesting practical issue for future study. A short discussion (as shown in line 350-354) may be helpful to improve the comprehensiveness of the paper.
Author Response
1. In section 4.2. line 257. The author calculates the bounding box of the object. However, it is not clear to me how they get the initial size of the object w_I and h_I. Are they extracted from the real-time image or estimated from pre-trained object database?
⦁ Thank you for your question. It is from real-time images.
⦁ This information has been added to section 4.2. line 239.
“First the width and height of the bounding box (wI , hI ) and object’s coordinate frame (OI) are extracted from the real-time image.”
2. As the proposed method is claimed as "enabling real-time identification", the speed or average time (at least estimated response time) of the algorithm to finish the task should be given.
⦁ Thank you for your comment. The Mask R-CNN inference speed is about ~29 fps. NVIDIA GeForce GTX 1080 Ti was used to run the model. The experimental device GPU of this paper is a GeForce RTX 2070 Mobile, and its average speed is 27.8 fps. The following section has been added to 6.3. Limitations and future work
⦁ The completion time of object classification and semantic segmentation of the implemented model is less than 0.5 seconds, and with the collision prediction algorithm running in tandem, the total response time is estimated to be 0.8 seconds.
3. In figure 8. i guess the bicycle rack is not able to be detected successfully, is it? It does not affect the novelty and efficiency of the method proposed. However, it maybe can be considered a limitation of current version of the algorithm or interesting practical issue for future study. A short discussion (as shown in line 350-354) may be helpful to improve the comprehensiveness of the paper.
⦁ Thank you for the comment. Yes, as you correctly pointed out, the model could misclassify objects that are overlayed or partially visible, such as the bike rack which is positioned almost within a bicycle. This is an inherent problem in any object classification and segmentation problems, and highly depends on the training dataset, particularly the number of images of a class (an object) that captures as many varieties and diversity as possible (e.g., camera angle, distance, ambient light, etc.). A section has been added to 6.1 to explain this aspect.
⦁ 6.1 Object Detection and Semntic Segmentation
“The inability to successfully detect the bicycle rack in Figure 8 could be attributed to a lack of diversity in the dataset used for training the model. If the model was trained on a limited set of bicycle rack images that do not cover various angles, lighting conditions, and designs, it may struggle to detect an object that is partially blocked or overlapped by other objects or showing from an angle at which the key features are not visible.”
Reviewer 3 Report
The paper is well written and structured. The authors developed a system utilizing vision transformers and multimodal feedback modules to aid navigation and collision avoidance for visually impaired. The introduction is relevant and theory based. The methods are generally appropriate. Clarification of a few details should be provided.
1. What is object detection and semantic segmentation performance? Please report corresponding IoU scores.
2. How is collision prediction model evaluated? While the authors used Fig. 9 to demonstrate, there is no accuracies or other evaluation metrics reported.
3. Fig. 9 demonstrated collision prediction in which a tall person is walking toward the user. What if a kid is walking toward the user? Will it make a difference? What’s the variations of the samples used for testing?
4. Table 3: What’s included in vegetation? What’s the difference between vegetation and tree?
5. What’s the limitation of the proposed method?
The quality of English is good.
Author Response
1. What is object detection and semantic segmentation performance? Please report corresponding IoU scores.
⦁ Thank you for your question. We used a Self-Supervised Learning model and based on our explanation on section 2.2. ‘Self-Supervised Learning (SSL), has shown its efficacy of learning discriminative feature representations for image classification, eliminating the requirement for manual annotation on labels.’ there was no need to annotate the data. Due to the absence of segmentation maps and bounding boxes in our dataset, we employed a Mask R-CNN model that had been fine-tuned on both the COCO and ADE20K datasets, as mentioned in Section 6.1. MIoU for semantic segmentation was 50, and the average precision for object detection was 51 [1].
2. How is collision prediction model evaluated? While the authors used Fig. 9 to demonstrate, there is no accuracies or other evaluation metrics reported.
⦁ The collision prediction algorithm was tested on a few scenarios but there were no metrics we considered to validate the accuracy of the algorithm. The goal was to demonstrate a proof of concept, and it is our next step to design and conduct functionality validation and usability test of the collision prediction algorithm. This is indeed one of the limitations of this paper and we will focus on addressing this point in our subsequent publication with formal evaluations done to demonstrate its performance quantitatively.
3. Fig. 9 demonstrated collision prediction in which a tall person is walking toward the user. What if a kid is walking toward the user? Will it make a difference? What’s the variations of the samples used for testing?
⦁ Thank you for your question. The model shows robustness when dealing with objects of different sizes, indicating its ability to handle diverse scale variations effectively. However, the model's performance has not been tested or evaluated specifically for subjects who are children. To illustrate the model's robustness in handling size variations, we conducted experiments where we used the same types of objects but with different sizes. In these tests, the object detection model successfully identified and detected all the objects, regardless of their varying dimensions. Additionally, we conducted real-time testing to evaluate the model's performance in dynamic scenarios. During this testing, the user was walking, leading to changes in the sizes of individuals with different heights as the user moved closer to or farther away from them. The model's ability to consistently detect people despite these changing sizes was confirmed through these experiments.
⦁ We’ve added this information to section 6.2. Collision Prediction (pg 14)
⦁ “The model shows robustness when dealing with objects of different sizes, indicating its ability to handle diverse scale variations effectively”
4. Table 3: What’s included in vegetation? What’s the difference between vegetation and trees?
⦁ Thank you for your question. In the context of this study, when referring to "vegetation," we specifically pertain to the plants found on the ground including bushes, herbs and shrubs. However, it should be noted that the "tree" class encompasses only the trees themselves.
5. What’s the limitation of the proposed method?
⦁ Thank you for your question. We’ve revised section 6.3 to address the limitations and future work.

Round 2
Reviewer 1 Report
I am grateful for providing appropriate answers to my questions. Thank you